# Associations Between Hemoglobin and Serum Iron Levels and the Risk of Mortality Among Patients with Coronary Artery Disease

**DOI:** 10.3390/nu17010139

**Published:** 2024-12-31

**Authors:** Qing Li, Zhijun Pan, Yupeng Zeng, Xu Wang, Dan Li, Ting Yin, Qian Chen, Wenhua Ling

**Affiliations:** 1School of Public Health, Sun Yat-sen University, 74, Zhongshan Rd. 2, Guangzhou 510080, China; liqing99@mail.sysu.edu.cn (Q.L.); panzhj5@mail2.sysu.edu.cn (Z.P.); zengyp27@mail2.sysu.edu.cn (Y.Z.); lidan58@mail.sysu.edu.cn (D.L.); 2Department of Clinical Nutrition, Guangdong Provincial People’s Hospital (Guangdong Academy of Medical Sciences), Southern Medical University, 106, Zhongshan Rd. 2, Guangzhou 510080, China; wangxu@gdph.org.cn; 3School of Public Health, Ningxia Medical University, 1160 Shengli Rd., Yinchuan 750004, China; 20160108@nxmu.edu.cn; 4Department of Cardiology, Sun Yat-sen Memorial Hospital, Sun Yat-sen University, 107 Yanjiang West Rd., Guangzhou 510120, China

**Keywords:** coronary artery disease, iron, hemoglobin, mortality, prospective cohort study

## Abstract

Background: This study aimed to investigate the relationship between hemoglobin and serum iron levels and mortality risk in patients with coronary artery disease (CAD). Methods: We analyzed data from 3224 patients with CAD using Cox proportional hazards regression models to assess the association of hemoglobin and serum iron levels with cardiovascular and all-cause mortality from the Guangdong coronary artery disease cohort. Results: Over a median follow-up period of 8.9 years, 636 patients died, including 403 from cardiovascular causes. Higher hemoglobin and serum iron levels were linked to a reduced risk of cardiovascular and all-cause mortality. Patients in the highest quartiles of hemoglobin and serum iron levels had multivariable-adjusted hazard ratios (HRs) of 0.62 (95% CI, 0.46–0.85) and 0.51 (95% CI, 0.37–0.69) for cardiovascular mortality and 0.64 (95% CI, 0.50–0.83) and 0.67 (95% CI, 0.53–0.85) for all-cause mortality, compared with those in the lowest quartile. A one-standard-deviation increase in hemoglobin and serum iron levels corresponded to a 19% and 24% reduction in cardiovascular mortality risk and a 19% reduction in all-cause mortality risk for both factors. Restricted cubic spline analysis revealed L-shaped and U-shaped associations between hemoglobin and serum iron levels and cardiovascular and all-cause mortality, respectively. Conclusions: Hemoglobin and serum iron levels were significantly associated with lower risks of cardiovascular and all-cause mortality in patients with CAD. Further research is needed to evaluate the effects of iron supplementation in these patients.

## 1. Introduction

Iron is a vital trace element found predominantly in hemoglobin and is essential for oxygen transport in red blood cells. Maintaining adequate iron levels is crucial for preventing anemia, which can result from insufficient dietary intake or inflammation that impairs iron absorption [1]. Conversely, iron overload due to abnormal erythropoiesis or genetic disruptions in iron homeostasis can increase the risk of organ damage and infections, significantly affecting liver and heart function [2]. Disturbances of iron metabolism may have deleterious consequences in severe pathological conditions such as cardiovascular diseases (CVDs), diabetes, cancer, and neurodegenerative diseases. It has been widely documented that aging is associated with dyshomeostasis of iron metabolism and regulation in both rodents and humans [3]. With aging, there is an incremental acquisition of several CVD risk factors during an individual’s lifespan, making age a significant contributor to cardiovascular risk [4]. Anemia, a condition closely related to impaired iron metabolism, is particularly common in older adult individuals and contributes significantly to morbidity and mortality in this population [5]. The interplay between aging, anemia, and cardiovascular risk highlights the importance of exploring the roles of hemoglobin and serum iron levels in determining mortality outcomes in older adult patients with CVD.

While the role of iron in red blood cell formation is well established, its involvement in other aspects of blood cell biology and broader metabolic processes is still emerging. Beyond erythropoiesis, iron plays key roles in oxygen and lipid metabolism, protein synthesis, and DNA replication [1]. Serum iron levels, along with transferrin, total iron-binding capacity (TIBC), and transferrin saturation, provide insights into an individual’s iron status, whereas ferritin concentrations primarily indicate iron stores [6]. The “iron hypothesis”, introduced by Sullivan in the early 1980s, suggested that elevated body iron levels might increase the risk of coronary artery disease (CAD), based on the observation of reduced CAD prevalence in populations with lower iron stores [7]. However, subsequent research on the relationship between iron status and CAD mortality has yielded inconclusive results [8,9]. Some studies, such as those by Corti et al. [8] suggest that higher serum iron levels may have a protective effect against mortality. However, others like Grammer et al. [9] have found that low hemoglobin and iron depletion are associated with an increased risk of CAD. Iron deficiency has been linked to adverse cardiovascular outcomes. For example, Parikh et al. [10] found an association between low hemoglobin levels and cardiovascular mortality in heart failure patients. Conversely, Bagheri et al. [11] reported high serum iron levels in patients with severe CAD, underscoring the complexity of iron’s impact on cardiovascular health. Further investigations by Salonen et al. [12] and Magnusson et al. [13] examined iron overload and its correlation with CAD, yielding mixed conclusions about the roles of TIBC and ferritin in CAD. Despite numerous studies, including meta-analyses [14], the association between iron parameters and CAD risk remains unclear, highlighting a significant gap in current understanding.

Despite these conflicting findings, iron homeostasis is recognized as a critical determinant of heart function, with both deficiency and overload linked to CAD and heart failure [15,16]. The potential toxicity of iron in CAD patients remains contentious, with hypotheses suggesting that oxidative stress, catalytic oxidation of low-density lipoprotein particles, and excessive reactive oxygen species production may contribute to vessel wall damage and atherosclerosis [1,17,18].

Studies investigating the association of iron status markers with CAD risk have been limited by conflicting results and the complexity of iron metabolism. A single marker may provide inadequate or misleading information. Therefore, this study aimed to explore the association between hemoglobin, serum iron, transferrin, TIBC, transferrin saturation, ferritin, and mortality risk among CAD patients recruited from the Guangdong coronary artery disease cohort (GCADC). This paper will examine how variations in these iron-related parameters influence mortality rates among patients diagnosed with CAD, contributing valuable insights into managing iron levels in this patient population.

## 2. Methods

### 2.1. Study Population

The GCADC is a prospective study of individuals aged 40 to 85 years from Guangdong province. The detailed cohort design and methodology have been previously published [19]. We initially recruited 1977 patients diagnosed with CAD from three hospitals between 2008 and 2011 using standardized selection, inclusion, and diagnostic criteria. In 2014, we expanded the cohort by identifying an additional 1622 CAD patients from hospital electronic medical records who were not previously included, bringing the total number of recruited CAD patients in the GCADC to 3599. All patients were diagnosed with CAD according to the International Classification of Diseases, 10th revision (codes I20–I25). After excluding 375 subjects with missing data including serum iron, hemoglobin, and follow-up date, we included 3224 CAD subjects in the primary analysis, focusing on serum iron and hemoglobin levels. Supplementary subgroup analyses were conducted with available data on serum ferritin, transferrin, and TIBC for 1190, 1473, and 1470 patients, respectively. Written informed consent was obtained from the initial 1977 participants, and anonymous data from electronic medical records were used for the 2014 recruits. The research protocol was approved by the Ethics Committee of School of Public Health, Sun Yat-sen University (approval code: 2015[019], approval date: 3 March 2015), and all clinical investigations followed the principles of the Helsinki Declaration.

### 2.2. Clinical Data Collection and Biological Measurements

Demographic information, smoking and alcohol consumption habits, medical histories, and additional risk factors were collected through a standardized questionnaire administered during face-to-face interviews or extracted from electronic hospitalization records. Height and weight were measured using a height and weight scale with standard accuracy and sensitivity. Blood pressure was measured with a mercurial sphygmomanometer while the patient was stable. After an overnight fast of at least 12 h post-admission, venous blood samples were collected and stored at −80 °C. Biochemical variables, including serum iron, hemoglobin, transferrin, TIBC, triglycerides (TG), total cholesterol (TC), low-density lipoprotein cholesterol (LDL-C), high-density lipoprotein cholesterol (HDL-C), fasting plasma glucose (FPG), and creatinine levels, were measured using a Hitachi 7600-020 Automatic Analyzer (Hitachi High-Technologies Corporation, Tokyo, Japan) following clinical laboratory guidelines. Plasma ferritin levels were measured via electrochemiluminescence immunoassay using the Architect i4000 system (Abbott Laboratories, Abbott Park, IL, USA). C-reactive protein (CRP) was measured using flow cytometry. Estimated glomerular filtration rates (eGFRs) were calculated to assess renal function and were categorized as <60 mL/min per 1.73 m^2^ or ≥60 mL/min per 1.73 m^2^.

### 2.3. Prospective Follow-Up

The primary outcomes were cardiovascular and all-cause mortality. Follow-up data were collected from hospital records, telephone conversations with patients or their family members, and the death registration system at the Guangdong Provincial Centers for Disease Control and Prevention. The follow-up period was conducted annually from the date of the blood sample collection to the date of death or until 30 November 2019.

### 2.4. Statistical Analyses

Continuous variables with normal distributions are presented as means and standard deviations, while non-normally distributed variables are presented as medians with 25th and 75th percentiles. Categorical data are represented as numbers and percentages. Analysis of variance was used for continuous normally distributed data, and log-transformation was applied to non-normally distributed continuous data prior to analysis. Categorical variables were analyzed using the chi-square test. Serum iron and hemoglobin were analyzed as continuous variables and categorized into quartiles by sex. Spearman’s correlation and partial correlation coefficients were used to analyze the relationships between serum iron, hemoglobin, and cardiovascular risk factors. Cox proportional hazard models evaluated the associations between serum iron (μmol/L), hemoglobin (g/L), and the risks of cardiovascular and all-cause mortality. We adjusted for a comprehensive set of confounders in our analysis, including demographic variables (age (years) and sex (male/female)), lifestyle factors (smoking status (never, past, or current), alcohol consumption (never, past, or current)), clinical characteristics (BMI (kg/m^2^), systolic and diastolic blood pressure (mmHg), non-HDL-C (mmol/L), and TG (mmol/L)), comorbidities (diabetes measured by FPG (mmol/L), renal function measured by eGFR (mL/min per 1.73 m^2^, %), and inflammation indicated by CRP (mg/L)), and medication use (anti-diabetic, anti-hypertensive, anti-platelet, and cholesterol-lowering drugs (yes/no)). CAD-specific factors, including the CAD type (acute and chronic) and duration of CAD (year), were considered to account for their potential influence on outcomes. The models also included the adjusted hemoglobin (except for serum iron) and serum iron (except for hemoglobin) levels. Subgroup analyses were conducted based on variables such as age, type of CAD, sex, BMI (≥24 or <24), and history of diabetes. Statistical analyses were performed using IBM^®^ SPSS^®^ software, version 22.0, with a two-tailed *p*-value < 0.05 considered statistically significant.

## 3. Results

### 3.1. Baseline Characteristics

The cohort consisted of 3224 patients (63.6% male) diagnosed with CAD, with a median follow-up duration of 8.9 years. Table 1 summarizes the baseline characteristics. Males had higher levels of serum iron, hemoglobin, and CRP, as well as higher rates of smoking, alcohol use, and acute coronary syndrome, compared with females. Females had higher mean ages, SBP, TC, LDL-C, HDL-C, and TG; a longer history of CAD; and a higher prevalence of comorbidities such as reduced eGFR (<60 mL/min/1.73 m^2^), along with more frequent use of antihypertensive and antidiabetic medications.

Serum iron levels were positively correlated with hemoglobin, BMI, SBP, TC, LDL-C, HDL-C, and TG levels (all *p* < 0.05) and negatively correlated with sex, age, CRP, and FPG (all *p* < 0.05). These correlations remained after adjusting for age, sex, and BMI, except for TG. Hemoglobin levels also positively correlated with BMI, TC, LDL-C, and TG and inversely with sex, age, CRP, and FPG (all *p* < 0.05), with stronger correlations after adjustment (Appendix A).

### 3.2. Hemoglobin Concentrations and Mortality Risks

During the follow-up period, 636 all-cause deaths and 403 cardiovascular deaths were recorded. Multivariable adjustment (model 3) revealed hazard ratios (HRs) for cardiovascular mortality of 0.64 (95% confidence interval [CI], 0.50–0.83), 0.58 (95% CI, 0.44–0.77), and 0.62 (95% CI, 0.46–0.85) for the second, third, and fourth quartiles of hemoglobin levels, respectively. Similarly, for all-cause mortality, HRs were 0.64 (95% CI, 0.52–0.79), 0.66 (95% CI, 0.53–0.82), and 0.64 (95% CI, 0.50–0.83) for the second, third, and fourth quartiles, respectively (Table 2). For each one-standard-deviation (SD) increase in hemoglobin concentrations, the HRs for cardiovascular and all-cause mortality risks were 0.81 (95% CI, 0.73–0.90) and 0.81 (95% CI, 0.74–0.88), respectively. Restricted cubic splines in Cox models showed that low hemoglobin levels (<130 g/L) were significantly associated with higher mortality risk (Figure 1). These results indicated that higher hemoglobin levels were associated with reduced risks of cardiovascular and all-cause mortality among CAD patients.

Multivariable subgroup analyses, stratified by age, type of CAD, sex, BMI, and history of diabetes, demonstrated consistent associations between hemoglobin levels and both cardiovascular and all-cause mortality. However, a significant interaction between hemoglobin levels and CAD type was found for all-cause mortality, as well as between hemoglobin levels and a history of diabetes for both cardiovascular and all-cause mortality. No statistically significant interaction was found between hemoglobin levels and other variables (Appendix A).

### 3.3. Serum Iron Concentrations and Mortality Risks

Following the multivariable adjustment in model 3, the HRs for cardiovascular mortality across the four quartiles of serum iron levels were 1.00, 0.72 (95% CI, 0.56–0.92), 0.44 (95% CI, 0.33–0.59), and 0.51 (95% CI, 0.37–0.69). Similarly, the HRs for all-cause mortality were 1.00, 0.84 (95% CI, 0.69–1.03), 0.51 (95% CI, 0.40–0.64), and 0.67 (95% CI, 0.53–0.85) (Table 3). For each one-SD increase in serum iron levels, there was a 24% reduction in the risk of cardiovascular mortality and a 19% reduction in the risk of all-cause mortality (all *p* < 0.05) (Table 3). Notably, low serum iron levels were associated with increased mortality risks, as indicated by the U-shaped curves observed in the Cox models (Figure 2). Separate analyses demonstrated that high iron levels had an independent protective effect on the mortality risk, regardless of hemoglobin levels (Appendix A).

Subgroup analyses, stratified by age, CAD type, sex, BMI, and history of diabetes, confirmed consistent associations between serum iron levels and mortality risks, with the exception of an interaction with diabetes history affecting cardiovascular mortality risk (Appendix A). Specifically, in CAD patients with diabetes, each one-SD increase in serum iron levels corresponded to a 37% reduction in cardiovascular mortality risk and a 29% reduction in all-cause mortality risk (Appendix A).

### 3.4. Serum Ferritin and Mortality Risks

After multivariable adjustments in model 3, each one-SD increase in plasma ferritin levels was associated with a 25% increase in cardiovascular mortality risk and a 21% increase in all-cause mortality risk (all *p* < 0.05), as presented in Appendix A. HRs for individuals in the highest versus lowest quartile of serum ferritin levels were 1.55 (95% CI, 1.04–2.31) for cardiovascular mortality and 1.45 (95% CI, 1.05–2.01) for all-cause mortality. Restricted cubic splines in Cox models also showed that higher plasma ferritin levels were associated with higher mortality risk (Appendix A).

### 3.5. Serum Transferrin and Mortality Risks

Multivariable adjustments revealed U-shaped associations between serum transferrin levels and both cardiovascular and all-cause mortality among CAD patients, as illustrated in Appendix A. The third quartile of serum transferrin levels exhibited the lowest HRs for cardiovascular mortality (HR: 0.51; 95% CI: 0.30–0.86) and all-cause mortality (HR: 0.66; 95% CI: 0.44–0.98) compared with the first quartile, as documented in Appendix A. These findings suggest an optimal range of transferrin levels that may confer protective effects against mortality in this patient population.

### 3.6. TIBC and Mortality Risks

The restricted cubic splines in Cox models, after multivariable adjustment, demonstrated U-shaped associations between TIBC levels and both cardiovascular and all-cause mortality among CAD patients, as depicted in Appendix A. The third quartile of TIBC levels showed the lowest hazard ratios for cardiovascular mortality (HR: 0.59; 95% CI: 0.35–0.99) and all-cause mortality (HR: 0.66; 95% CI: 0.45–0.99) compared with the first quartile, as noted in Appendix A. This suggests an optimal range of TIBC that may offer protection against mortality in CAD patients.

### 3.7. Transferrin Saturation and Mortality Risks

Each one-SD increase in transferrin saturation was associated with a 17% reduction in cardiovascular mortality risk and a 21% reduction in all-cause mortality risk in model 3 (all *p* < 0.05), as detailed in Appendix A. This suggests that an optimal range of transferrin saturation may provide protective effects against mortality risks in CAD patients. However, the observed U-shaped curve implies that excessively high or low transferrin saturation levels could be detrimental, highlighting the importance of maintaining iron levels within a moderate range (30–40%), as shown in Appendix A.

## 4. Discussion

Our study indicated that among Chinese patients with CAD, hemoglobin and serum iron levels higher than the first quartile were associated with significantly lower risks of cardiovascular and all-cause mortality. The observed U-shaped associations suggested that both iron deficiency and overload could negatively impact cardiovascular health. Additionally, we found that transferrin saturation levels in the moderate range of 30–40% were associated with the lowest risk of mortality, with deviations from this range linked to higher mortality. In contrast, plasma ferritin levels higher than the first quartile were associated with higher risks of cardiovascular and all-cause mortality among CAD patients after adjusting for other cardiovascular-associated biomarkers. Hence, iron status markers (hemoglobin, serum iron, transferrin, transferrin saturation, TIBC, and ferritin) may serve as independent predictors of mortality risk among patients with CAD.

Previous epidemiological studies investigating the association of hemoglobin and CAD have yielded conflicting results regarding associations with CVD [9,20] and mortality [21,22,23]. The UK Biobank study observed a J-shaped association between hemoglobin levels and CAD risk among European adults, where both extremely low and high hemoglobin levels were detrimental. Additionally, Mendelian randomization showed a linear positive correlation, indicating an 8% increase in CAD risk per standard deviation increase in genetically predicted hemoglobin levels [20]. The Ludwigshafen Risk and Cardiovascular Health Study showed that low hemoglobin levels were independently associated with CAD [9] but had only a marginal association with cardiovascular and total mortality in patients with stable CAD [23]. Low or high hemoglobin concentrations were associated with elevated cardiovascular and all-cause mortality [22], emphasizing the importance of maintaining hemoglobin levels within the normal range. However, a large cohort study of 39,922 patients with acute coronary syndrome found a highly statistically significant and independent association between low hemoglobin levels and adverse cardiovascular outcomes [21]. This is consistent with our findings that low hemoglobin is an independent predictor of cardiovascular and all-cause mortality in CAD patients. This relationship may be partially explained by the prevalence of diabetes and renal dysfunction in CAD patients. The relationship between diabetes and CAD has been extensively documented, with CVD representing the most common cause of death in diabetic patients [24].

In addition to hemoglobin concentrations, the present study also evaluated the associations of iron status biomarkers with mortality. Our findings support the beneficial effects of higher serum iron and hemoglobin levels on reducing mortality risks in CAD patients. Mendelian randomization analysis suggests a protective effect of higher iron status against CAD [25], specifically indicating that increased levels of serum iron, transferrin saturation, and ferritin are associated with a reduced risk of CAD. Conversely, higher levels of transferrin, which indicate lower iron status, appear to increase CAD risk [25]. Similarly, Corti et al. [8] found that higher serum iron levels were linked to lower all-cause and cardiovascular mortality in older adults, aligning with our findings on the risks associated with iron levels. These observations are supported by Jankowska et al. [26], who noted that optimal transferrin saturation levels could discriminate iron deficiency and protect against heart disease. Furthermore, the research by Grammer et al. [9] and Kiechl et al. [27] suggests that maintaining adequate iron status is crucial for preventing the progression of atherosclerosis and improving cardiovascular outcomes. Moreover, our observation of moderate transferrin saturation levels (30–40%) offering the lowest risk of mortality is supported by Guedes et al. [28], who found similar beneficial effects at a 40% transferrin saturation level for all-cause mortality and major adverse cardiovascular events. Recent studies, such as that by Zhang et al. [29], discuss the dual nature of iron in the cardiovascular system, emphasizing that both deficiency and overload can significantly impact heart health, particularly in heart failure patients. This aligns with our observations of a beneficial range for transferrin saturation and the detrimental effects of iron overload indicated by high plasma ferritin levels [29]. Moreover, research by Lupu et al. [30] explores the therapeutic implications of iron modulation in CVD, suggesting that controlled iron supplementation could offer protective benefits in chronic cardiovascular conditions, thus supporting our findings of reduced mortality with higher serum iron and hemoglobin levels [30]. However, these findings contrast with the results of a meta-analysis that indicated no significant association between serum iron, TIBC, serum ferritin, and CAD but found a significant negative association between transferrin saturation and CAD [31]. The meta-analysis compared the top and bottom tertiles of body iron status, which may have masked the different effects of mild iron deficiency and mild iron overload on CAD risk [31].

The protective role of moderate transferrin saturation observed in our study may be due to its ability to maintain a balance in iron availability. This balance ensures sufficient iron for erythropoiesis and other cellular functions while avoiding toxic levels. Excessive iron can catalyze the production of reactive oxygen species, exacerbating oxidative stress. This, in turn, can lead to endothelial dysfunction, a critical factor in the pathogenesis of atherosclerosis [27,32]. By avoiding both iron deficiency and overload, moderate transferrin saturation might optimize cellular function and minimize oxidative stress.

Furthermore, the relationship between higher ferritin levels and increased mortality could be explained by ferritin’s role as an acute-phase reactant. Elevated ferritin levels might reflect not only increased iron stores but also systemic inflammation, which is a known risk factor for atherosclerosis and subsequent cardiovascular events [33,34]. This inflammation can lead to increased vascular instability and plaque vulnerability, contributing to higher mortality rates.

The observed discrepancies could stem from methodological differences in measuring and defining iron status across studies; variations in population characteristics like age, sex, and baseline health conditions; and the interaction of iron with other risk factors such as inflammation, diabetes, and lipid profiles. These factors necessitate a cautious interpretation of the relationship between iron status and cardiovascular health as the influence of iron may differ substantially based on individual and demographic factors.

The field of cardiovascular research presents varied findings regarding the impact of iron status on heart disease, illustrating the complexity of iron metabolism and its interaction with cardiovascular health. Some studies suggest a role for iron in promoting atherosclerosis, where elevated iron stores may contribute to the formation of atherosclerotic plaques [27,35]. Conversely, Tuomainen et al. [18] indicated a potential oxidative role of iron in CVD, which could explain the increased cardiovascular risk with higher iron levels. Further, studies found no significant relationship between serum ferritin and cardiovascular events, challenging the hypothesis that iron overload heightens cardiovascular risk [36,37]. It is important to manage iron metabolism to prevent dysregulation that can lead to severe cardiovascular conditions, which may arise from both iron deficiency and overload [38]. More recent studies found no conclusive evidence linking iron status with CAD [14,39], suggesting that factors like sex, population genetics, dietary influences, or methodological differences across studies may play roles in these varied outcomes.

Iron metabolism plays a complex role in cardiovascular health, influencing processes ranging from oxygen transport and mitochondrial function to inflammation and oxidative stress [40,41]. Moreover, the U-shaped curves observed for iron and transferrin saturation suggest that both deficiency and excess of iron could be harmful, emphasizing the need for a delicate balance in iron homeostasis [7,42]. This balance is crucial as iron overload can lead to toxic effects, including the promotion of lipid peroxidation and foam cell formation, both key processes in atherogenesis [35,43]. In summary, the associations between iron metabolism markers and cardiovascular outcomes likely involve a complex interplay of metabolic, oxidative, and inflammatory pathways, each contributing to the overall cardiovascular risk profile in individuals with CAD. The interplay between iron metabolism and cardiovascular health is intricate and multifaceted. Maintaining appropriate iron levels within a specific therapeutic window may be key to optimizing treatment strategies for CAD as both iron deficiency and overload pose significant risks. Future research should continue to explore these dynamics, particularly in diverse populations and under varying health conditions to better understand and exploit iron’s potential in CVD management.

A significant strength of our study is the comprehensive evaluation of multiple iron markers and their association with mortality in a large cohort of patients with CAD. Our analysis of the independent effects of hemoglobin, serum iron, transferrin, transferrin saturation, TIBC, and ferritin on mortality provides valuable insights into the complex interactions between iron metabolism and cardiovascular health. Furthermore, utilizing a well-defined cohort with long-term follow-up enhances the reliability of our findings. However, our study does have several limitations. The observational nature of our research precludes drawing causal inferences between iron status and mortality. Additionally, our cohort was hospital-based, potentially limiting the generalizability of our findings to the broader population. The absence of data on iron supplementation and dietary iron intake could also have influenced our results. Another potential limitation is the influence of confounding factors such as inflammation and chronic disease states, which can affect ferritin levels independently of iron status. Addressing these confounding factors in future research could help clarify the relationships between iron parameters and cardiovascular outcomes.

## 5. Conclusions

Our study highlights that hemoglobin, serum iron, transferrin, transferrin saturation, TIBC, and ferritin are strongly associated with mortality risks in patients with CAD. These markers likely reflect broader iron status and its physiological effects, suggesting that the root cause may be iron dysregulation rather than the biomarkers themselves. Higher levels of hemoglobin and serum iron were associated with lower mortality risks, suggesting the potential importance of maintaining balanced iron homeostasis in this population. Based on these findings, clinicians may consider regular monitoring of hemoglobin and serum iron levels in CAD patients as part of their routine management. Future research should explore the potential benefits of interventions aimed at optimizing iron levels, such as targeted supplementation, while avoiding the risks of iron overload. Further large-scale cohort studies and randomized controlled trials are needed to validate and expand upon these findings, particularly in diverse populations and with a focus on the long-term effects of iron modulation.

## Figures and Tables

**Figure 1 nutrients-17-00139-f001:**
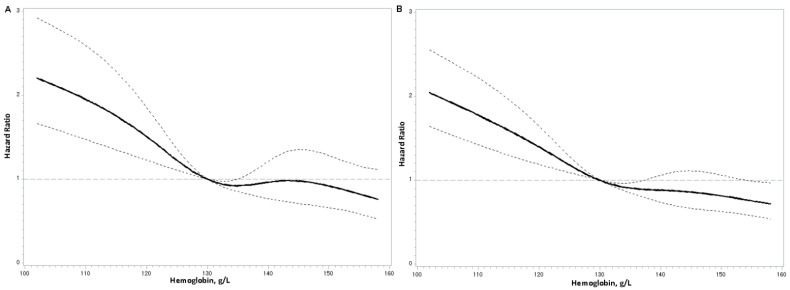
Restricted cubic splines in Cox models of hemoglobin levels with cardiovascular (**A**) and all-cause (**B**) mortality risk. Adjustments were made for age; sex; alcohol drinking status; smoking status; body mass index; systolic blood pressure; diastolic blood pressure; fasting plasma glucose; non-high-density lipoprotein cholesterol; triglycerides; duration of coronary artery disease; estimated glomerular filtration rate; type of coronary artery disease (acute and chronic); C-reactive protein; the use of anti-diabetic, anti-platelet, cholesterol-lowering, and anti-hypertensive drugs; and serum iron levels. The middle solid line represents the hazard ratio, while the upper and lower dashed lines indicate the 95% confidence interval.

**Figure 2 nutrients-17-00139-f002:**
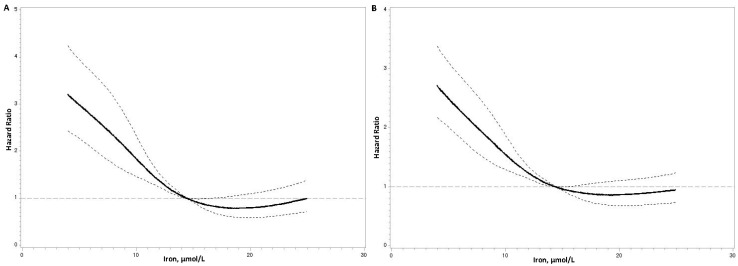
Restricted cubic splines in Cox models of serum iron levels with cardiovascular (**A**) and all-cause (**B**) mortality risk. Adjustments were made for age; sex; alcohol drinking status; smoking status; body mass index; systolic blood pressure; diastolic blood pressure; fasting plasma glucose; non-high-density lipoprotein cholesterol; triglycerides; duration of coronary artery disease; estimated glomerular filtration rate; type of coronary artery disease (acute and chronic); C-reactive protein; the use of anti-diabetic, anti-platelet, cholesterol-lowering, and anti-hypertensive drugs; and hemoglobin levels. The middle solid line represents the hazard ratio, while the upper and lower dashed lines indicate the 95% confidence interval.

**Table 1 nutrients-17-00139-t001:** Baseline characteristics of subjects with coronary artery disease according to sex.

Variables	Total (*n* = 3224)	Male (*n* = 2052)	Female (*n* = 1172)	*p* Value
Serum iron (μmol/L)	13.1 (9.1–17.5)	13.2 (9.2–18.2)	12.9 (9.1–16.5)	0.001
Hemoglobin (g/L)	133.0 (121.0–143.0)	138.0 (127.0–147.0)	125.0 (115.0–133.0)	<0.001
Age, year	64.2 (11.1)	62.4 (11.6)	67.3 (9.5)	<0.001
Body mass index, kg/m^2^	24.0 (3.4)	24.0 (3.2)	24.0 (3.6)	0.968
Systolic blood pressure, mmHg	135.4 (22.4)	133.4 (22.4)	138.7 (22.0)	<0.001
Diastolic blood pressure, mmHg	78.1 (12.7)	78.3 (13.1)	77.8 (12.0)	0.301
Fasting plasma glucose, mmol/L	6.62 (2.84)	6.55 (2.61)	6.74 (3.20)	0.060
Total cholesterol, mmol/L	4.80 (1.15)	4.65 (1.12)	5.05 (1.16)	<0.001
Low-density lipoprotein cholesterol, mmol/L	2.99 (0.98)	2.93 (0.97)	3.09 (1.00)	<0.001
High-density lipoprotein cholesterol, mmol/L	1.14 (0.31)	1.08 (0.28)	1.25 (0.33)	<0.001
Triglycerides, mmol/L	1.53 (1.09–2.17)	1.51 (1.07–2.14)	1.56 (1.12–2.21)	0.036
C-reactive protein *, mg/L	3.11 (1.11–11.16)	3.39 (1.11–13.28)	2.70 (1.10–8.26)	<0.001
Estimated glomerular filtration rate, mL/min per 1.73 m^2^, %				<0.001
<60	28.7	24.5	35.8	
≥60	71.3	75.5	64.2	
Duration of coronary artery disease, year				
First diagnosed coronary artery disease (*n* = 2114)	-	-	-	-
History of coronary artery disease (*n* = 1110)	3.00 (0.86–7.16)	2.34 (0.73–6.34)	4.00 (1.00–9.80)	<0.001
Smoking status, %				<0.001
Never	60.8	41.2	95.2	
Current	29.1	43.9	3.2	
Past	10.1	15.0	1.6	
Alcohol drinking status, %				<0.001
Never	83.9	75.3	98.7	
Current	11.7	17.8	1.1	
Past	4.4	6.8	0.2	
Type of coronary artery disease, %				<0.001
Acute coronary syndrome (*n* = 1823)	56.5	61.5	47.9	
Chronic coronary artery disease (*n* = 1401)	43.5	38.5	52.1	
History of diseases (yes), %				
Hypertension	76.2	74.5	79.4	0.002
Dyslipidemia	19.3	18.7	20.4	0.232
Diabetes	23.7	21.7	27.0	0.001
Use of medication before admission (Yes), %				
Anti-hypertensive drugs	48.6	44.2	57.2	<0.001
Cholesterol-lowering drugs	12.3	12.5	11.8	0.668
Anti-diabetic drugs	16.7	14.7	20.5	0.002
Anti-platelet drugs	19.9	20.7	18.2	0.202

Data are presented as mean (standard deviation) for continuous variables, median [25th, 75th] for non-normally distributed variables, or percent for categorical variables. * Log transformed before analysis.

**Table 2 nutrients-17-00139-t002:** Hazard ratios of cardiovascular and all-cause mortality according to baseline hemoglobin levels.

	Hemoglobin Levels						
	Quartile 1	Quartile 2	Quartile 3	Quartile 4	*p* Value	Hazard Ratio per 1 SD Increment	*p* Value
Cardiovascular mortality							
N	816	819	799	790			
Person-years	7.86	8.44	8.60	8.78			
Number of deaths	143	104	80	76			
Model 1	1.00	0.61 (0.47–0.78)	0.50 (0.38–0.66)	0.51 (0.38–0.69)	<0.001	0.74 (0.68–0.82)	<0.001
Model 2	1.00	0.61 (0.47–0.79)	0.52 (0.39–0.69)	0.53 (0.39–0.72)	<0.001	0.76 (0.68–0.84)	<0.001
Model 3	1.00	0.64 (0.50–0.83)	0.58 (0.44–0.77)	0.62 (0.46–0.85)	<0.001	0.81 (0.73–0.90)	<0.001
All-cause mortality							
N	816	819	799	790			
Person-years	7.86	8.44	8.60	8.78			
Number of deaths	228	161	134	113			
Model 1	1.00	0.59 (0.48–0.72)	0.55 (0.45–0.69)	0.51 (0.41–0.65)	<0.001	0.74 (0.68–0.79)	<0.001
Model 2	1.00	0.61 (0.50–0.75)	0.60 (0.48–0.75)	0.56 (0.44–0.72)	<0.001	0.77 (0.71–0.83)	<0.001
Model 3	1.00	0.64 (0.52–0.79)	0.66 (0.53–0.82)	0.64 (0.50–0.83)	<0.001	0.81 (0.74–0.88)	<0.001

Model 1 was adjusted for age, sex, alcohol drinking status, and smoking status. Model 2 was adjusted for the model 1 covariates plus body mass index; systolic blood pressure; diastolic blood pressure; fasting plasma glucose; non-high-density lipoprotein cholesterol; triglycerides; duration of coronary artery disease; estimated glomerular filtration rate; type of coronary artery disease (acute and chronic); C-reactive protein; and the use of anti-diabetic, anti-platelet, cholesterol-lowering, and anti-hypertensive drugs. Model 3 was adjusted for model 2 covariates plus serum iron levels.

**Table 3 nutrients-17-00139-t003:** Hazard ratios of cardiovascular and all-cause mortality according to baseline serum iron levels.

	Serum Iron Levels						
	Quartile 1	Quartile 2	Quartile 3	Quartile 4	*p* Value	Hazard Ratio per 1 SD Increment	*p* Value
Cardiovascular mortality							
N	811	807	802	804			
Person-years	7.82	8.41	8.81	8.63			
Number of deaths	169	105	58	71			
Model 1	1.00	0.63 (0.50–0.81)	0.37 (0.27–0.49)	0.40 (0.29–0.53)	<0.001	0.67 (0.60–0.75)	<0.001
Model 2	1.00	0.68 (0.53–0.87)	0.41 (0.30–0.55)	0.46 (0.34–0.62)	<0.001	0.72 (0.64–0.81)	<0.001
Model 3	1.00	0.72 (0.56–0.92)	0.44 (0.33–0.59)	0.51 (0.37–0.69)	<0.001	0.76 (0.67–0.85)	<0.001
All–cause mortality							
N	811	807	802	804			
Person-years	7.82	8.41	8.81	8.63			
Number of deaths	241	174	99	122			
Model 1	1.00	0.74 (0.61–0.90)	0.42 (0.33–0.53)	0.51 (0.41–0.64)	<0.001	0.72 (0.65–0.78)	<0.001
Model 2	1.00	0.79 (0.65–0.97)	0.47 (0.37–0.59)	0.59 (0.47–0.75)	<0.001	0.77 (0.70–0.84)	<0.001
Model 3	1.00	0.84 (0.69–1.03)	0.51 (0.40–0.64)	0.67 (0.53–0.85)	<0.001	0.81 (0.74–0.89)	<0.001

Model 1 was adjusted for age, sex, alcohol drinking status, and smoking status. Model 2 was adjusted for model 1 covariates plus body mass index; systolic blood pressure; diastolic blood pressure; fasting plasma glucose; non-high-density lipoprotein cholesterol; triglycerides; duration of coronary artery disease; estimated glomerular filtration rate; type of coronary artery disease (acute and chronic); C-reactive protein; and the use of anti-diabetic, anti-platelet, cholesterol-lowering, and anti-hypertensive drugs. Model 3 was adjusted for model 2 covariates plus hemoglobin levels.

## Data Availability

The data presented in this study are available on request from the corresponding author due to ethical reasons.

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
