# Peer review of "Associations Between Hemoglobin and Serum Iron Levels and the Risk of Mortality Among Patients with Coronary Artery Disease"

_nutrients, 2024, doi:10.3390/nu17010139_

Round 1
Reviewer 1 Report
Comments and Suggestions for Authors
General Comment
Thank you for the opportunity to review this article. While the topic is highly relevant, the manuscript requires substantial improvements across all sections before it can be considered for publication.
Abstract
This section requires further development. Additional details about the methods should be included for clarity. The results should be presented in a more concise and focused manner.
1. Background: The study’s aim should be explicitly stated in this section.
2. Methods: Please provide more information about the study population. From where was the sample obtained? Was this a sub-analysis of an existing cohort? If so, specify the cohort. Include details about the main variables: How were hemoglobin and iron levels assessed? How was mortality determined?
3. Results: This section should be more concise. Consider focusing on the multivariate analysis results, with a brief summary of the other findings.
Introduction
While this section effectively highlights the importance of the research topic (mortality in CAD patients) and the role of iron levels, improvements in grammar and the use of connectors are necessary. Additional references should be included to support specific claims. Detailed comments are as follows:
1. Inter-paragraph spacing should be eliminated throughout the manuscript.
2. Line 60: If discussing multiple studies, more than one reference is required. Please check and correct accordingly.
3. Line 61: The use of "In contrast" may not be appropriate. The statement about high iron levels being protective and low levels increasing CAD risk seems complementary rather than contradictory.
4. Line 71: Consider removing the word "definitive" to enhance objectivity.
5. Lines 83–85: Consider incorporate the study population into the aim.
Methods
This section requires additional detail, particularly regarding the main variables and analytical methods.
1. Line 101: Provide more information about the data missing from excluded participants. Specify the variables affected.
2. Line 105: Correct the reference to Figure 1, as it does not appear to be the flowchart.
3. Lines 115–116: Provide specific details about the protocols used to measure height, weight, and blood pressure.
4. Lines 129–134: Clarify the process for registering mortality. How were mortality causes classified? What was the follow-up period? These details are critical for study reproducibility.
5. Statistical Analysis:
Separate the description of descriptive and multivariate analyses.
Provide details about the models and adjusted variables, including categories or definitions for each variable (e.g., age (in years), sex (male/female)).
Justify the selection of variables (e.g. based on literature or established risk factors).
Explain the rationale and how you performed for the "1-SD increment" analysis.
Did you examine other common causes of mortality, such as cancer?
Have you tested for trends (p-trend) in these associations?
Results
This section would benefit from improved clarity and focus on the key findings.
1. Lines 176–188: Please, focus on the results for the highest versus lowest quartiles, providing numeric results only. Apply this approach consistently throughout the results section.
2. Tables 2 and 3:
Improve the table structure.
Clearly specify the follow-up period in years.
Present ‘all-cause mortality’ at the top, followed by cardiovascular mortality.
Above each model, include the following information for each quartile: n (%), number of deaths, and person-years.
3. Moderate the use of causal language. Replace terms such as "increased" and "decreased" with "higher" or "lower" to avoid implying causation (e.g., lines 186 and 266).
Discussion
This section is well-structured and effectively contextualizes the findings within the existing literature.
1. Ensure consistent use of "CVD" for all references to "cardiovascular."
2. Lines 322–329: The first sentence is too long. Consider splitting it into shorter sentences for better readability.
Conclusion
Consider adding specific recommendations based on the study’s results, while clearly acknowledging the limitations. Suggest directions for future research, such as larger cohort studies or intervention trials to validate and expand upon these findings.
Author Response
Responses to Reviewer 1’s comments:
Abstract
This section requires further development. Additional details about the methods should be included for clarity. The results should be presented in a more concise and focused manner.
Re: We greatly appreciate your careful review and comments regarding our manuscript.
- Background: The study’s aim should be explicitly stated in this section.
Re: Thanks for the valuable comment. We have explicitly stated the study’s aim in this section as follows:
This study aims to investigate the relationship between hemoglobin and serum iron levels and mortality risk in patients with coronary artery disease (CAD).
- Methods: Please provide more information about the study population. From where was the sample obtained? Was this a sub-analysis of an existing cohort? If so, specify the cohort. Include details about the main variables: How were hemoglobin and iron levels assessed? How was mortality determined?
Re: Thanks for the valuable comment. Our sample was obtained from Guangdong Coronary Artery Disease Cohort, a prospective study of CAD patients recruited from multiple hospitals in Guangdong province. Serum iron and hemoglobin were measured using a Hitachi 7600-020 Automatic Analyzer following clinical laboratory guidelines. Mortality data were collected from hospital records, telephone conversations with patients or their family members, and the death registration system at the Guangdong Provincial Centers for Disease Control and Prevention.
We have provided more information about the study population in this section as follows:
Methods: We analyzed data from 3,224 patients with CAD using Cox proportional hazards regression models to assess the association of hemoglobin and serum iron levels with cardiovascular and all-cause mortality from the Guangdong Coronary Artery Disease Cohort.
- Results: This section should be more concise. Consider focusing on the multivariate analysis results, with a brief summary of the other findings.
Re: Thanks for the valuable comment. We have focused on the key findings from the Cox proportional hazards regression models analyses, with a concise summary of other findings to maintain focus.
Introduction
While this section effectively highlights the importance of the research topic (mortality in CAD patients) and the role of iron levels, improvements in grammar and the use of connectors are necessary. Additional references should be included to support specific claims. Detailed comments are as follows:
- Inter-paragraph spacing should be eliminated throughout the manuscript.
Re: Thanks for the valuable comment. We have eliminated the inter-paragraph spacing throughout the manuscript.
- Line 60: If discussing multiple studies, more than one reference is required. Please check and correct accordingly.
Re: Thank you for your observation. We fully agree that multiple references are needed when discussing multiple studies. We have added the appropriate additional references to ensure that all studies mentioned are properly cited.
- Line 61: The use of "In contrast" may not be appropriate. The statement about high iron levels being protective and low levels increasing CAD risk seems complementary rather than contradictory.
Re: Thanks for the valuable comment. We have changed "In contrast" to “However”.
- Line 71: Consider removing the word "definitive" to enhance objectivity.
Re: Thanks for the valuable comment. We have removed the word "definitive".
- Lines 83–85: Consider incorporate the study population into the aim.
Re: Thank you for your valuable suggestion. We agree that incorporating the study population into the aim will enhance the clarity and focus of the manuscript. We have revised the text accordingly to address your comment.
Methods
This section requires additional detail, particularly regarding the main variables and analytical methods.
- Line 101: Provide more information about the data missing from excluded participants. Specify the variables affected.
Re: Thanks for the valuable comment. We have provided variable information about the missing data as follows:
After excluding 375 subjects with missing data including serum iron, hemoglobin, and follow-up date, we included 3,224 CAD subjects in the primary analysis, focusing on serum iron and hemoglobin levels.
- Line 105: Correct the reference to Figure 1, as it does not appear to be the flowchart.
Re: Thanks for the valuable comment. We have deleted the reference to Figure 1.
- Lines 115–116: Provide specific details about the protocols used to measure height, weight, and blood pressure.
Re: Thanks for the valuable comment. We have provided specific details about the protocols used to measure height, weight, and blood pressure as follows:
Height and weight were measured using a height and weight scale with standard accuracy and sensitivity. Blood pressure was measured with a mercurial sphygmomanometer while the patient was stable.
- Lines 129–134: Clarify the process for registering mortality. How were mortality causes classified? What was the follow-up period? These details are critical for study reproducibility.
Re: Thanks for the valuable comment. Mortality causes were classified from hospital records, telephone conversations with patients or their family members, and the death registration system at the Guangdong Provincial Centers for Disease Control and Prevention. The follow-up period conducted annually from the date of blood sample collection to the date of death or until November 30, 2019.
- Statistical Analysis:
Separate the description of descriptive and multivariate analyses.
Re: Thanks for the comment. We have described them separately.
Provide details about the models and adjusted variables, including categories or definitions for each variable (e.g., age (in years), sex (male/female)).
Re: Thanks for the valuable comment. We have provided details about the models and adjusted variables as you suggested.
Justify the selection of variables (e.g. based on literature or established risk factors).
Re: Thanks for the valuable comment. We selected variables based on literature and known cardiovascular disease risk factors.
Explain the rationale and how you performed for the "1-SD increment" analysis.
Re: Thanks for the valuable comment. We calculated the standard deviation (SD) for hemoglobin and serum iron levels within our cohort. The SD reflects the variability of these biomarkers and allows for a uniform scaling of their effects. Standardized Variable = (Observed Value−Mean)/ Standard Deviation.
Using Cox proportional hazards regression models, we included the standardized variables (in units of 1-SD) as continuous predictors to estimate their associations with cardiovascular and all-cause mortality. This approach provided hazard ratios (HRs) corresponding to a 1-SD increment, independent of the original unit of measurement. The HRs derived from 1-SD increments represent the relative risk of mortality associated with a typical variation (1-SD) in hemoglobin or serum iron levels, making it easier to understand the clinical significance of their variability.
Did you examine other common causes of mortality, such as cancer?
Re: Thank you for your insightful question. Our study population consisted exclusively of patients diagnosed with coronary artery disease, and the primary focus of our research was on cardiovascular and all-cause mortality. This focus aligns with the main clinical and scientific objectives of understanding the association between hemoglobin and serum iron levels and mortality risk in this specific population. While cancer mortality is an important endpoint in other populations, but the number of cancer-related deaths was relatively small in our cohort, limiting the statistical power to perform meaningful analyses or derive robust conclusions regarding cancer-related mortality.
Have you tested for trends (p-trend) in these associations?
Re: Thank you for the valuable question. Yes, we tested for trends (p-trend) in the associations between hemoglobin and serum iron levels and mortality outcomes in our analysis. We assigned integers (1, 2, 3, 4) to the quartiles of hemoglobin and serum iron levels and included these ordinal variables in the regression models. The p-value for the coefficient of the ordinal variable was interpreted as the p-trend, indicating whether there was a statistically significant trend across quartiles.
Results
This section would benefit from improved clarity and focus on the key findings.
- Lines 176–188: Please, focus on the results for the highest versus lowest quartiles, providing numeric results only. Apply this approach consistently throughout the results section.
Re: Thank you for your thoughtful suggestion. In our study, we presented results for all four quartiles to provide a comprehensive understanding of the trends and to allow readers to observe the consistency of the associations across quartiles. This approach aligns with common practices in the literature, where presenting all quartiles helps illustrate dose-response relationships and ensures transparency in reporting.
We referred to multiple studies in the field, where the results across all quartiles are typically displayed to highlight incremental effects and trends. This comprehensive presentation allows readers to understand the progression of risk across the entire range of biomarker levels.
By showing results for all quartiles, including the corresponding 95% confidence intervals (CIs), we provided additional clarity regarding the statistical significance of the findings. The inclusion of 95% CIs highlights whether the associations are significant and allows readers to evaluate the precision of the hazard ratio estimates.
While we presented results for all quartiles, we ensured that comparisons between the highest and lowest quartiles were clearly emphasized in the text, as they represent the extremes of exposure and are often of greatest clinical and scientific relevance.
- Tables 2 and 3:
Improve the table structure.
Clearly specify the follow-up period in years.
Present ‘all-cause mortality’ at the top, followed by cardiovascular mortality.
Above each model, include the following information for each quartile: n (%), number of deaths, and person-years.
Re: Thanks for the valuable comment. We have modified Table 2 and Table 3 according to your suggestion.
- Moderate the use of causal language. Replace terms such as "increased" and "decreased" with "higher" or "lower" to avoid implying causation (e.g., lines 186 and 266).
Re: Thanks for the valuable comment. We have revised the paper according to your suggestion.
Discussion
This section is well-structured and effectively contextualizes the findings within the existing literature.
- Ensure consistent use of "CVD" for all references to "cardiovascular."
Re: Thank you for your helpful comment. We acknowledge that "CVD" is the abbreviation for "cardiovascular disease," and we carefully reviewed the manuscript to ensure that its usage aligns with the intended context. Specifically:
For references to cardiovascular disease, we have consistently used the abbreviation "CVD" throughout the text.
For instances where "cardiovascular" refers to a broader concept, such as "cardiovascular mortality," "cardiovascular risk," or "cardiovascular outcomes," we have retained the full term "cardiovascular" to avoid any potential misunderstanding.
- Lines 322–329: The first sentence is too long. Consider splitting it into shorter sentences for better readability.
Re: Thank you for your helpful comment. We have changed the sentence as follows:
The protective role of moderate transferrin saturation observed in our study may be due to its ability to maintain a balance in iron availability. his balance ensures sufficient iron for erythropoiesis and other cellular functions while avoiding toxic levels. Excessive iron can catalyze the production of reactive oxygen species, exacerbating oxidative stress. This, in turn, can lead to endothelial dysfunction, a critical factor in the pathogenesis of atherosclerosis.
Conclusion
Consider adding specific recommendations based on the study’s results, while clearly acknowledging the limitations. Suggest directions for future research, such as larger cohort studies or intervention trials to validate and expand upon these findings.
Re: Thank you for this insightful suggestion. We have revised the Conclusion section to incorporate specific recommendations based on the study’s findings. These include the potential clinical value of monitoring hemoglobin and serum iron levels in CAD patients and exploring targeted interventions to optimize iron levels. Additionally, we have explicitly acknowledged the study's limitations, such as its observational nature and lack of dietary iron data, which restrict causal interpretations. Lastly, we highlighted the need for further research, including larger cohort studies and intervention trials, to confirm and extend our findings in broader populations.
Reviewer 2 Report
Comments and Suggestions for Authors
Very interesting, practical and applicable study most particularly while providing useful information concerning optimal ranges of Fe parameters.
Introduction
The introduction could be completed since only mention low and high Fe levels in CAD
Line 52 – please change to ferritin concentrations.
It would be interesting to mention somewhere in the introduction the relation between aging and iron dyshomeostasis when considering age as a cardiovascular risk factor; anemia in elderly could also be considered.
“Disturbances of iron metabolism may have deleterious consequences in severe pathological conditions such as cardiovascular diseases, diabetes, cancer and neurodegenerative diseases. It has been widely documented that aging is associated with dyshomeostasis of iron metabolism and regulation in both rodents and humans” in Xu J, Jia Z, Knutson MD, Leeuwenburgh C. Impaired iron status in aging research. Int J Mol Sci. 2012;13(2):2368-2386. doi: 10.3390/ijms13022368. Epub 2012 Feb 22. PMID: 22408459; PMCID: PMC3292028, is an example.
“With aging, there is an incremental acquisition of several CVD risk factors in an individual’s lifespan” in Dhingra R, Vasan RS. Age as a risk factor. Med Clin North Am. 2012 Jan;96(1):87-91. doi: 10.1016/j.mcna.2011.11.003. Epub 2011 Dec 12. PMID: 22391253; PMCID: PMC3297980, is also an example.
Considering the results in line 185 “Restricted cubic splines in Cox models showed that low hemoglobin levels (< 130 g/L) were significantly associated with increased mortality risk”, a mention the of the relation between anemia and elderly whould improve the paper. “Anemia in the elderly (defined as people aged > 65 years) is common and increasing as the population ages. In older patients, anemia of any degree contributes significantly to morbidity and mortality” in Stauder R, Thein SL. Anemia in the elderly: clinical implications and new therapeutic concepts. Haematologica. 2014 Jul;99(7):1127-30. doi: 10.3324/haematol.2014.109967. PMID: 24986873; PMCID: PMC4077071, is another example.
Methods
Study population
A control group is not explicitly mentioned. If people without CAD are not included, the control grup is the CAD group without Fe supplementation? If so, the complete paradigm f the paper should be changed – indeed what the authors are studying is the impact of Fe supplementation in CVD.
Line 120 – Please explain briefly the reason why fasting plasma glucose (FPG), and creatinine levels were measured; somewhere in the paper the authors should mention the relationship between diabetes and CAD. “A close link exists between DM and cardiovascular disease (CVD), which is the most prevalent cause of morbidity and mortality in diabetic patients” in Leon BM, Maddox TM. Diabetes and cardiovascular disease: Epidemiology, biological mechanisms, treatment recommendations and future research. World J Diabetes. 2015 Oct 10;6(13):1246-58. doi: 10.4239/wjd.v6.i13.1246. PMID: 26468341; PMCID: PMC4600176., is an example. The same rationale for renal function.
The parameters which where considered as confounders should be described more explicitly.
Results
Line 226 – “higher plasma ferritin levels were associated with higher mortality risk”, perhaps too much Fe outside the optimal range, when considering a U-shaped curve?
Discussion
Line 260 - Where is written “higher levels of hemoglobin and serum iron”, the term higher is not explicit. I believe the authors do not intent to mention the highest levels , when considering the detrimental effects of excessive Fe. Therefore, “higher” should referred as “higher than ….” In line 266, the same rational for the word moderate should be applied; the same for “high plasma ferritin”.
Line 268- “iron status markers may serve as independent predictors of mortality risk among patients with CAD”, which ones exactly? All the studied ones?
Conclusion
Where is written, “Our study highlights the significant impact of hemoglobin, serum iron, transferrin, transferrin saturation, TIBC, and ferritin on mortality risks in patients with CAD”, the authors could wonder if this parameters can itself have an impact on CAD mortality risk, or if since they are biomarkers of Fe status, the root of the question is not simply Fe status.
Author Response
Responses to Reviewer 2’s comments:
Introduction
The introduction could be completed since only mention low and high Fe levels in CAD
Re: We greatly appreciate your careful review and comments regarding our manuscript.
Line 52 – please change to ferritin concentrations.
Re: Thank you for your suggestion. We have changed “ferritin levels” to “ferritin concentrations”.
It would be interesting to mention somewhere in the introduction the relation between aging and iron dyshomeostasis when considering age as a cardiovascular risk factor; anemia in elderly could also be considered.
“Disturbances of iron metabolism may have deleterious consequences in severe pathological conditions such as cardiovascular diseases, diabetes, cancer and neurodegenerative diseases. It has been widely documented that aging is associated with dyshomeostasis of iron metabolism and regulation in both rodents and humans” in Xu J, Jia Z, Knutson MD, Leeuwenburgh C. Impaired iron status in aging research. Int J Mol Sci. 2012;13(2):2368-2386. doi: 10.3390/ijms13022368 Add to Citavi project by DOI. Epub 2012 Feb 22. PMID: 22408459 Add to Citavi project by Pubmed ID; PMCID: PMC3292028, Add to Citavi project by PMC ID is an example.
“With aging, there is an incremental acquisition of several CVD risk factors in an individual’s lifespan” in Dhingra R, Vasan RS. Age as a risk factor. Med Clin North Am. 2012 Jan;96(1):87-91. doi: 10.1016/j.mcna.2011.11.003 Add to Citavi project by DOI. Epub 2011 Dec 12. PMID: 22391253 Add to Citavi project by Pubmed ID; PMCID: PMC3297980, Add to Citavi project by PMC ID is also an example.
Considering the results in line 185 “Restricted cubic splines in Cox models showed that low hemoglobin levels (< 130 g/L) were significantly associated with increased mortality risk”, a mention the of the relation between anemia and elderly whould improve the paper. “Anemia in the elderly (defined as people aged > 65 years) is common and increasing as the population ages. In older patients, anemia of any degree contributes significantly to morbidity and mortality” in Stauder R, Thein SL. Anemia in the elderly: clinical implications and new therapeutic concepts. Haematologica. 2014 Jul;99(7):1127-30. doi: 10.3324/haematol.2014.109967 Add to Citavi project by DOI. PMID: 24986873 Add to Citavi project by Pubmed ID; PMCID: PMC4077071, Add to Citavi project by PMC ID is another example.
Re: Thank you for this insightful suggestion. We have revised the Introduction to include discussions on the relationship between aging and iron dyshomeostasis, as well as the association of anemia with morbidity and mortality in elderly populations. We have cited relevant studies (Xu et al., 2012; Dhingra and Vasan, 2012; Stauder and Thein, 2014) to provide context and support for these additions. These revisions strengthen the rationale for our study by highlighting the interplay between aging, anemia, and cardiovascular risk, particularly in the context of hemoglobin and serum iron levels.
Methods
Study population
A control group is not explicitly mentioned. If people without CAD are not included, the control grup is the CAD group without Fe supplementation? If so, the complete paradigm f the paper should be changed – indeed what the authors are studying is the impact of Fe supplementation in CVD.
Re: Thank you for raising this important question. We would like to clarify that our study is a prospective cohort study, and no intervention (including iron supplementation) was administered as part of the research. All participants were diagnosed with coronary artery disease (CAD) at baseline, and the primary aim of the study was to evaluate the association between hemoglobin and serum iron levels and mortality outcomes (both cardiovascular and all-cause mortality).
As all participants in this study had CAD, there was no control group consisting of individuals without CAD. Instead, we investigated variability within the CAD population by stratifying individuals based on their baseline hemoglobin and serum iron levels.
The study did not involve any iron supplementation or other interventions. Thus, it is not designed to study the impact of iron supplementation on cardiovascular outcomes. Instead, it is an observational study that analyzes naturally occurring variations in hemoglobin and serum iron levels among CAD patients and their associations with mortality outcomes.
The focus of our study is on the prognostic implications of hemoglobin and serum iron levels in CAD patients, with death as the primary outcome. This approach allows us to explore the potential role of these biomarkers in predicting mortality risks in a high-risk population without any external intervention.
Line 120 – Please explain briefly the reason why fasting plasma glucose (FPG), and creatinine levels were measured; somewhere in the paper the authors should mention the relationship between diabetes and CAD. “A close link exists between DM and cardiovascular disease (CVD), which is the most prevalent cause of morbidity and mortality in diabetic patients” in Leon BM, Maddox TM. Diabetes and cardiovascular disease: Epidemiology, biological mechanisms, treatment recommendations and future research. World J Diabetes. 2015 Oct 10;6(13):1246-58. doi: 10.4239/wjd.v6.i13.1246 Add to Citavi project by DOI. PMID: 26468341 Add to Citavi project by Pubmed ID; PMCID: PMC4600176. Add to Citavi project by PMC ID, is an example. The same rationale for renal function.
Re: Thank you for your insightful comment. We have clarified the rationale for measuring fasting plasma glucose (FPG) and creatinine levels in the study, as well as highlighted the relationship between diabetes and coronary artery disease (CAD). Below is the additional explanation and the relevant revisions to the manuscript.
Fasting Plasma Glucose (FPG):
FPG was measured to assess glycemic status and identify diabetes mellitus (DM), which is a major risk factor for CAD. There is a well-established link between DM and cardiovascular disease (CVD), as CVD represents the most prevalent cause of morbidity and mortality in diabetic patients (Leon and Maddox, 2015). Measuring FPG allows for the adjustment of glycemic status as a confounding variable in our analysis, thereby enhancing the validity of the associations between hemoglobin, serum iron, and mortality outcomes in CAD patients.
Creatinine Levels:
Serum creatinine was measured to evaluate renal function, which is closely associated with both CAD and mortality risk. Renal dysfunction can contribute to anemia and iron metabolism disorders, further influencing cardiovascular outcomes. Adjusting for creatinine levels in our models ensures that the observed relationships between hemoglobin, serum iron, and mortality risks are not confounded by renal dysfunction.
The parameters which where considered as confounders should be described more explicitly.
Re: Thank you for your valuable comments. We have described the confounding factors in the paper more explicitly.
Results
Line 226 – “higher plasma ferritin levels were associated with higher mortality risk”, perhaps too much Fe outside the optimal range, when considering a U-shaped curve?
Re: Thank you for your insightful observation. In the context of plasma ferritin, elevated levels might reflect excessive iron stores that can catalyze the production of reactive oxygen species, leading to oxidative stress, endothelial dysfunction, and subsequent cardiovascular damage. Additionally, ferritin is an acute-phase reactant, and elevated levels may also indicate underlying inflammation, which is independently associated with increased cardiovascular and all-cause mortality. Thus, both iron overload and inflammation might contribute to the observed association.
Discussion
Line 260 - Where is written “higher levels of hemoglobin and serum iron”, the term higher is not explicit. I believe the authors do not intent to mention the highest levels , when considering the detrimental effects of excessive Fe. Therefore, “higher” should referred as “higher than ….” In line 266, the same rational for the word moderate should be applied; the same for “high plasma ferritin”.
Re: Thank you for pointing out the need for clarity in our terminology regarding “higher,” “moderate,” and “high” levels of iron-related markers. We agree that these terms should be explicitly defined to avoid misinterpretation, especially given the detrimental effects of excessive iron levels. To address your concerns, we have revised the manuscript.
Line 268- “iron status markers may serve as independent predictors of mortality risk among patients with CAD”, which ones exactly? All the studied ones?
Re: Thank you for your valuable comment regarding the specific iron status markers that may serve as independent predictors of mortality risk among patients with coronary artery disease (CAD). In our study, we examined six markers of iron status: hemoglobin, serum iron, transferrin, transferrin saturation, total iron-binding capacity (TIBC), and ferritin. Among the studied markers, hemoglobin, serum iron, transferrin, transferrin saturation, TIBC, and ferritin were identified as independent predictors of mortality risk among patients with CAD, with each exhibiting distinct patterns of association with mortality.
We have made the following modifications in the paper: Hence, iron status markers (hemoglobin, serum iron, transferrin, transferrin saturation, TIBC, and ferritin) may serve as independent predictors of mortality risk among patients with CAD.
Conclusion
Where is written, “Our study highlights the significant impact of hemoglobin, serum iron, transferrin, transferrin saturation, TIBC, and ferritin on mortality risks in patients with CAD”, the authors could wonder if this parameters can itself have an impact on CAD mortality risk, or if since they are biomarkers of Fe status, the root of the question is not simply Fe status.
Re: Thank you for your insightful comment. You are correct that the iron-related biomarkers we studied—hemoglobin, serum iron, transferrin, transferrin saturation, TIBC, and ferritin—are indicative of iron status, and their associations with mortality risks could reflect underlying iron homeostasis rather than direct causative effects of the biomarkers themselves.
To address this point, we acknowledge that these biomarkers primarily serve as indicators of the broader iron status and its physiological consequences, including oxidative stress, inflammation, and other mechanisms impacting cardiovascular health. While our study found significant associations between these markers and CAD mortality risk, further investigation is needed to delineate whether these relationships are direct or mediated by systemic iron dysregulation.
To incorporate this clarification, we have revised the relevant sentence in the manuscript as follows: Our study highlights that hemoglobin, serum iron, transferrin, transferrin saturation, TIBC, and ferritin are strongly associated with mortality risks in patients with CAD. These markers likely reflect broader iron status and its physiological effects, suggesting that the root cause may be iron dysregulation rather than the biomarkers themselves.
Reviewer 3 Report
Comments and Suggestions for Authors
I congratulate the writers for their excellent data and explanations, but I also want to draw attention to a few details that should be considered when discussing risk factors. Some of them and their effects have already been discussed. For those of you who haven't, I advise you to bring up the study's shortcomings during the conversation.
Initial Demographic and Clinical Features
Body mass index (BMI), age, and sex.
Comorbidities (such as diabetes, high blood pressure, and chronic renal disease) are present. Drugs taken (such as statins, iron supplements, or antiplatelet medicines).
Levels of Hemoglobin
Since both extremes might affect cardiovascular outcomes, find the range of hemoglobin levels to distinguish between anemia (low hemoglobin) and polycythemia (high hemoglobin). Sort hemoglobin levels into groups that are clinically significant so they can be analyzed.
Serum Iron Levels
concentration of iron in the serum.
Transferrin saturation (TSAT) and total iron-binding capacity (TIBC). Ferritin, an acute-phase reactant, is used to measure inflammation and iron storage.
To distinguish between iron deficiency anemia and anemia of chronic disease (inflammatory etiology), use C-reactive protein (CRP) or other inflammatory markers.
Serum creatinine and estimated glomerular filtration rate (eGFR), since the results of CAD and anemia can be impacted by chronic renal disease.
Cardiac Function Specifications
Because anemia affects heart failure with reduced ejection fraction (HFrEF) and heart failure with preserved ejection fraction (HFpEF) differently, left ventricular ejection fraction (LVEF) is used to distinguish between the two types. Troponins and NT-proBNP.
Lifestyle and Nutritional Factors
onsumption of iron by diet and supplements. levels of physical activity, smoking, and alcohol use.
Analysis of Subgroups
Analyze relationships independently in subgroups, such as those based on age, sex, the severity of CAD, or the presence of heart failure.
By thoroughly examining these variables, you can gain a better understanding of how hemoglobin and serum iron levels affect the risk of death in patients with CAD, possibly revealing therapeutic targets or intervention thresholds.
Author Response
Responses to Reviewer 3’s comments:
Comments and Suggestions for Authors
I congratulate the writers for their excellent data and explanations, but I also want to draw attention to a few details that should be considered when discussing risk factors. Some of them and their effects have already been discussed. For those of you who haven't, I advise you to bring up the study's shortcomings during the conversation.
Re: Thank you for your encouraging feedback and for pointing out the importance of addressing the study's limitations when discussing risk factors.
Initial Demographic and Clinical Features
Body mass index (BMI), age, and sex.
Comorbidities (such as diabetes, high blood pressure, and chronic renal disease) are present. Drugs taken (such as statins, iron supplements, or antiplatelet medicines).
Re: Thank you for your valuable comment regarding the presentation of the initial demographic and clinical features of our study population. We have described the variables in detail in the method as follows:
We adjusted for a comprehensive set of confounders in our analysis, including demographic variables (age (years), sex (male/female)), lifestyle factors (smoking status(Never, Past, Current), alcohol consumption (Never, Past, Current)), clinical characteristics (BMI( kg/m2), systolic and diastolic blood pressure (mmHg), non-HDL-C (mmol/L), TG (mmol/L)), comorbidities ((diabetes measured by FPG (mmol/L), renal function measured by eGFR (ml/min per 1.73 m2, %)), inflammation indicated by CRP(mg/L)), and medication use (anti-diabetic, anti-hypertensive, anti-platelet, and cholesterol-lowering drugs (Yes/No)), CAD-specific factors, including the CAD type (acute and chronic) and duration of CAD (year), were considered to account for their potential influence on outcomes.
Levels of Hemoglobin
Since both extremes might affect cardiovascular outcomes, find the range of hemoglobin levels to distinguish between anemia (low hemoglobin) and polycythemia (high hemoglobin). Sort hemoglobin levels into groups that are clinically significant so they can be analyzed.
Re: Thank you for your thoughtful comment regarding the clinical relevance of hemoglobin levels and the importance of categorizing these into clinically significant groups. We agree that stratifying hemoglobin levels into clinically relevant ranges will enhance the clarity of our findings and their applicability to clinical practice. Our results showed that Restricted cubic splines in Cox models showed that low hemoglobin levels (< 130 g/L) were significantly associated with higher mortality risk.
Serum Iron Levels
concentration of iron in the serum.
Transferrin saturation (TSAT) and total iron-binding capacity (TIBC). Ferritin, an acute-phase reactant, is used to measure inflammation and iron storage.
Re: Thank you for your valuable comment. We fully agree with your perspective. In the manuscript, we have also described the dual role of ferritin as both an acute-phase reactant and a marker of iron storage in the discussion section.
To distinguish between iron deficiency anemia and anemia of chronic disease (inflammatory etiology), use C-reactive protein (CRP) or other inflammatory markers.
Re: Thank you for your insightful comment regarding the differentiation between iron deficiency anemia (IDA) and anemia of chronic disease (ACD) using inflammatory markers such as C-reactive protein (CRP). We fully agree with this approach and acknowledge its importance. In our study, we measured CRP levels to assess systemic inflammation. Elevated CRP levels were used to identify a pro-inflammatory state, which is characteristic of ACD. Additionally, we observed that ferritin, an acute-phase reactant, was elevated in the presence of inflammation, further supporting the differentiation between IDA and ACD. These markers provide valuable insights into the etiology of anemia and its implications for mortality risks.
Serum creatinine and estimated glomerular filtration rate (eGFR), since the results of CAD and anemia can be impacted by chronic renal disease.
Re: Thank you for highlighting the importance of serum creatinine and estimated glomerular filtration rate (eGFR) in the context of chronic renal disease, which can impact both CAD and anemia outcomes. We completely agree with your perspective. In our study, we adjusted for eGFR in the statistical analyses to account for its potential influence on the observed associations. Additionally, we have discussed the role of renal function in the context of CAD and anemia in the manuscript.
Cardiac Function Specifications
Because anemia affects heart failure with reduced ejection fraction (HFrEF) and heart failure with preserved ejection fraction (HFpEF) differently, left ventricular ejection fraction (LVEF) is used to distinguish between the two types. Troponins and NT-proBNP.
Re: Thank you for your insightful comment regarding the need to consider cardiac function specifications such as left ventricular ejection fraction to distinguish between heart failure with reduced ejection fraction and heart failure with preserved ejection fraction. We completely agree with your observation and acknowledge the importance of these parameters.
In our study, while we did not include left ventricular ejection fraction, troponins, or NT-proBNP as part of the data analysis, we recognize their potential to provide a more nuanced understanding of the interplay between anemia, cardiac function, and mortality. This represents a limitation of our study, and we have added this point to the discussion section.
Lifestyle and Nutritional Factors
consumption of iron by diet and supplements. levels of physical activity, smoking, and alcohol use.
Re: Thank you for your thoughtful comment regarding lifestyle and nutritional factors, including dietary iron intake, supplement use, physical activity levels, smoking, and alcohol use. We fully acknowledge that these factors can significantly impact iron status and related outcomes. In our study, while we collected data on smoking and alcohol use, we did not have detailed information on iron intake from diet or supplements. We recognize this as a limitation and have discussed it in the manuscript.
Analysis of Subgroups
Analyze relationships independently in subgroups, such as those based on age, sex, the severity of CAD, or the presence of heart failure.
Re: Thank you for your insightful comment regarding the analysis of relationships in specific subgroups. We completely agree with the importance of subgroup analysis to better understand the associations between iron status markers and outcomes.
In our study, we conducted subgroup analyses based on the following factors (Supplementary Table 2 and 3):
Age: Subgroup analyses were performed for younger (<65 years) and older (≥65 years) patients.
Type of CAD: Stratification by acute coronary syndrome (ACS) and chronic coronary artery disease was included.
Sex: Male and female subgroups were analyzed independently.
Body Mass Index (BMI): Patients were categorized into BMI ≥24 and <24 groups.
History of Diabetes: Subgroup analyses were conducted for patients with and without diabetes.
These stratified analyses demonstrated consistent associations across subgroups, with some interactions observed, such as between hemoglobin levels and the type of CAD or diabetes status.
By thoroughly examining these variables, you can gain a better understanding of how hemoglobin and serum iron levels affect the risk of death in patients with CAD, possibly revealing therapeutic targets or intervention thresholds.
Re: Thank you for your insightful comment. We fully agree that a thorough examination of variables such as hemoglobin and serum iron levels can provide critical insights into their roles in influencing mortality risk in patients with CAD.
Round 2
Reviewer 1 Report
Comments and Suggestions for Authors
A brief reminder for the authors: It would be helpful if, when submitting a revised version of the manuscript, you indicate the modifications made in red or using track changes. Otherwise, the reviewer may not be able to accurately identify the changes incorporated. Thank you.
Comment: Tables 2 and 3 remain incorrect. Please unify the format to match Table 1, ensuring the removal of margins. The n(%) and person-years should also be included for specific causes, such as cardiovascular mortality in your case.
Author Response
Responses to Reviewer 1’s comments:
A brief reminder for the authors: It would be helpful if, when submitting a revised version of the manuscript, you indicate the modifications made in red or using track changes. Otherwise, the reviewer may not be able to accurately identify the changes incorporated. Thank you.
Re: Thank you for your helpful reminder. We sincerely apologize for not providing a version with modifications clearly indicated in our previous submission. In this revised version, we have used track changes to ensure that all modifications made to the manuscript are visible and easy to identify. We hope this will facilitate your review process.
Comment: Tables 2 and 3 remain incorrect. Please unify the format to match Table 1, ensuring the removal of margins. The n(%) and person-years should also be included for specific causes, such as cardiovascular mortality in your case.
Re: Thank you for your valuable comments regarding Tables 2 and 3. We have made the requested changes to Tables 2 and 3 to match the format of Table 1, including unifying the layout and ensuring the removal of margins. Additionally, n(%) and person-years for specific causes have been included as your suggestions.